# A Digital Image Confidentiality Scheme Based on Pseudo-Quantum Chaos and Lucas Sequence

**DOI:** 10.3390/e22111276

**Published:** 2020-11-11

**Authors:** Khushbu Khalid Butt, Guohui Li, Fawad Masood, Sajid Khan

**Affiliations:** 1School of Computer Science and Technology, Huazhong University of Science and Technology, Wuhan 430074, China; khushbukhalid@hust.edu.cn (K.K.B.); khansajid@hust.edu.cn (S.K.); 2Faculty of Computer Science Department, Huazhong University of Science and Technology, Wuhan 430074, China; 3College of Information Engineering, Yangzhou University, Yangzhou 225127, China; Fawadkttk@gmail.com; 4Department of Electrical Engineering, Institute of Space Technology, Islamabad 44000, Pakistan

**Keywords:** quantum logistic map, image encryption, Lucas series, substitution box, statistical analysis

## Abstract

Several secure image encryption systems have been researched and formed by chaotic mechanisms in current decades. This work recommends an innovative quantum color image encryption method focused on the Lucas series-based substitution box to enhance the competence of encryption. The suggested encryption technique has more excellent key space and significant confidentiality. The chaotic system, along with the substitution box, exhibits additional complicated dynamical behavior, sufficient arbitrariness, and uncertainty than all others focused on just chaotic models. Theoretical and simulation assessments show that the offered image encryption performs admirably, its traditional equivalents in terms by efficiency in terms of statistical analysis.

## 1. Introduction

The advancement of computer networks directs to the extra efficient retrieval of digital images over multimedia networks. Encryption is used to secure sensitive information being transmitted across the web. A wide range of chaos behaviors is very hard to predict that are apparently random and unpredictable. Chaos theory defines the randomness behavior that exists in the chaotic complex system and it can be prescribed by utilizing mathematical models. Chaotic models are extensively employed to secure data because of its desired properties, including ergodicity, unpredictability, and sensitive dependence on initial conditions, the wrong initial condition will lead to non-chaotic behavior. These properties, particularly in scientific and engineering disciplines, have attracted vast attention, designing new cryptographic algorithms and cryptanalysis. A chaotic system’s dynamics exhibit fascinating nonlinear effects, leading to complete security and key space in data encryption. Chaos played a vital role in designing robust cryptosystems such as the construction of S-boxes, image encryption algorithms, random number generators, and so on figure [1,2,3,4,5,6,7]. Quantum chaos-based encrypted images will play an essential role in the future quantum computer era as a specific and crucial quantum information type. Several representation schemes or models for quantum images have been developed for various purposes. With the advent of time, concerns raised that if classical chaotic systems become quantized. The subject has become quantum chaos. This study is based on quantum versions of classical chaotic systems. The map-based on the chaotic quantum system gives deep insight into the nature of quantum chaos [8]. The quantized version of the classical chaotic map has better properties. The quantized version of the classical map (quantum map) may be thought of transformation based on quantum equivalents of canonical transformations. However, there is no unique procedure of quantizing the classical map. Many researchers and cryptographers have utilized quantum maps in the context of quantum chaos [9,10]. Classical chaotic systems can be distinguished due to its high sensitivity towards the initial condition. At the same time, quantum chaos depends upon the parameters and its sensitivity in the Hamiltonian in the subject of chaotic dynamics [11,12]. The interesting properties of quantum chaos can be applied in cryptography for the image encryption process—numerous works on designing cryptographic algorithms employing quantum chaos [13]. The existing schemes of quantum chaos have utilized the physical process of quantum chaos. We have developed a system merely based on equations that are widely used chaotic quantum map. Quantum communication transmission is of immense significant interest to research scientists, physicists, and mathematics. It is a methodology related to creating innovative quantum techniques/protocols for encoding, retrieving, and visual processing information [14]. It is expected to bring about a new era of scientific advances in computing, communication, machine learning, and cryptography because quantum computing can resolve the mismanagement of classical computers [15]. Quantum images will indeed perform a vital character in the quantum-based period, as a particular and essential type of quantum theory. A series of available versions or prototypes with quantum images have been developed for various purposes. Due to its prospective use in secure communication, quantum encryption process and substitution often play a significant role in diverse scientific and engineering fields. By using a chaotic quantum map, Lucas series with strong S-box based hybrid dynamical models, this article provides new approaches for encryption. Since coupled quantum logistic encryption, are itself ideal for quality encryption, the addition of Lucas series and substitution box stipulates highly secure encryption programmed [16,17,18,19,20,21]. Liang et al. [22] suggested a new quantum encryption technique dependent on affine transformation and logistic map controlled XOR image operations. In 2017, a quantum-based encryption approach was applied in the work of Zhou et al. [23] utilizing a hyper-chaotic method and reiterative Arnold transforms to manage image cycle shift operations. Ten et al. [24] offer quantum encryption method depending upon Chen’s hyper-chaotic system.

Considerable work has been carried out in recent years to prevent unauthorized users from accessing digital files. Zhi et al. [25] suggested an image confidentiality algorithm by merging a cat map along with Chen’s chaotic technique to scramble the image data. New hyper-chaos-based image data encryption techniques have been suggested by diffusing and scrambling images with hyper-chaos sequences [26,27,28,29]. Partial authentication is indicated in [30], which decreases the encipher and decipher time of video, image processing and data transmission. Even then, it is only appropriate for a partial compression procedure and is therefore not considered as a global standard. The diffusion and permutation features of the cellular automation (CA)-based image confidentiality system [31,32] is satisfactory to several strengths. A symmetrical data encryption technique focused on a two-dimensional conventional baker map is shown in [33]. An optimal image data encryption method dependent on permutation—diffusion and skew tent map structure has lately been proposed in [34]. In [35], Eslami and Bakhshandeh review the security vulnerabilities of [34] toward known-plaintext and chosen-plaintext attacks and reveal that the susceptibility of plain text, as observed by the authors, is not sufficient. It is recommended that the technique be emphasized more than twice to tolerate differential attacks.

The articles presented in the literature section, depending on the quantum or non-quantum mechanism requires a high computational cost and time. To reduce cost and time, we have offered a new approach to quantum image encryption. The suggested algorithm works as a hybrid model utilizing quantum chaotic logistic map at its initial stage for an image encryption process. This system possesses high randomness comparative to traditional techniques and classical chaotic maps. The increased complexity of quantum chaos generating random sequencing all depends upon the initial state and parameters that control the encryption process. Moreover, we employed the Lucas series to add more randomness to the previous sequences generated by quantum chaos phase. Finally, each pixel is substituted to add more diffusion to the scheme. The proposed algorithm is tested over several standard statistical tests. Performance analysis stage depicted that the proposed method is reliable and highly robust. 

The provided algorithm comprises two algorithms: The first algorithm is for diffusion by the implementation of a chaotic quantum system, and the second algorithm is to add confusion by the application of the substitution box generated by the Lucas series and pseudo-random number generator. Experimental outcomes, depending on different categories of data protection and pace performance, demonstrates that the suggested encryption system can manage trade-offs among security requirements and speed performance.

The remaining article is structured as follows. Section 2 delivers some fundamental knowledge of the suggested technique. In Section 3 and Section 4, we have presented the design of encryption and decryption techniques, respectively. Security review and discussion involving contrasts with other methods are delivered in Section 5. Finally, in last section we have included some conclusion remarks.

## 2. Fundamental Knowledge

### 2.1. Chaotic Logistic Classical Map

The classical logistic map gained much attention using the idea of chaos due to its simple representation and operation [36]. A chaotic logistic map possesses better dynamic characteristics as well as a uniform feature of invariant density. A simple interpretation of the chaos is that unpredictable systems are highly sensitive to its initial conditions. The main reason of such sensitivity to initial conditions is the repetitive recapitulation and lengthening of the given space described by the map. Mathematically the system is illustrated as [37]: (1)$$x_{n}=r(x_{n})\left(1-x_{n}\right)$$
where in the equation $x_{n}$ is the initial condition and *r* is the control parameter. The value of $x_{n}$ must befall in the range of 0 and 1, i.e., $x_{n}\in \left(0,1\right)$ whilst the value of r must be in the specified range of 3 and 4. The chaos region will emerged in the interval r∈(3.54, 1). If control parameter or initial conditions are changed from its specified range the system will show completely different attractor. The initial key must be correct for proper encryption and decryption process. 

### 2.2. Pseudo-Quantum Chaotic System

In 1990, Goggin et al. [38] derived a dissipative logistic map with quantum corrections method by coupling the quantum kicked to the bath of harmonic oscillators. The quasicontinuum model is introduced to describe the dissipation from the bath, and then analyzed the resulting expectation-value map by taking a truncated of expectation value. In order to study the quantum correlation effects they wrote $\hat{\alpha }=\langle \hat{\alpha }\rangle +\delta \hat{\alpha }$ where $\delta \hat{\alpha }$ depicts quantum fluctuations about the operator $\langle \hat{\alpha }\rangle$ and has the property: $\delta \hat{\alpha }\to 0$. They initiate a period-doubling route to the classical behavior as a dissipation parameter is enhanced and other fascinating aspects at transitional values of this parameter. In this way, they study what effects correlations of the form $\langle \delta {\hat{\alpha }}^{†}\delta \hat{\alpha }\rangle$, and, $\langle \delta \hat{\alpha }\delta \hat{\alpha }\rangle$, etc., have the coupling to the bath is varied. 

Considering the one-dimensional like classical logistic equation as:(2)$$\langle {\hat{\alpha }}_{i+1}\rangle =r\left(\langle {\hat{\alpha }}_{i}\rangle -\langle {\hat{\alpha }}_{i}{}^{†}{\hat{\alpha }}_{i}\rangle \right),$$

In the above Equation (2) *r* is a chaotic parameter that is adjustable where ${\hat{\alpha }}_{i},\;\mathrm{and}\;\langle {\hat{\alpha }}_{i}{}^{†}\rangle$ are the two annihilation and creation operators of the boson system of the bath. Now using the assumption for the aforementioned $\hat{\alpha }=\langle \hat{\alpha }\rangle +\delta \hat{\alpha }$ where $\delta \hat{\alpha }$ shows quantum fluctuations about $\langle \hat{\alpha }\rangle$ and has the property: $\delta \hat{\alpha }\to 0$. By taking a truncated system of expectation value and correlation in the manner of Goggin et al. can be written as: (3)$$\langle {\hat{\alpha }}_{i+1}\rangle =r\left(\langle {\hat{\alpha }}_{i}\rangle -{\left|\langle {\hat{\alpha }}_{i}\rangle \right|}^{2}\right)-r\langle \delta {\hat{\alpha }}_{i}{}^{†}\delta {\hat{\alpha }}_{i}\rangle ,$$

From the Heisenberg equation of motion $\delta {\hat{\alpha }}_{i}$, we can derive the equation for $\langle \delta {\hat{\alpha }}^{†}\delta \hat{\alpha }\rangle$ which gives the appearance of third-order quantum corrections. Now for higher-order correlations $\langle \delta {\hat{\alpha }}^{†}\delta \hat{\alpha }\rangle$, $\langle \delta \hat{\alpha }\delta \hat{\alpha }\rangle$ and their Hermitian conjugates are ignored, which results in the following set of equations:(4a)$$x_{i+1}\left(1\right)=r\left(x_{i}\left(1\right)-{\left|x_{i}\left(1\right)\right|}^{2}\right)-rx_{i}\left(2\right),$$
(4b)$$x_{i+1}\left(2\right)=-x_{i}\left(2\right)e^{-2\beta }+2re^{-\beta }\left[-x_{i}\left(1\right)x_{i}\left(2\right)-x_{i}\left(1\right)x_{i}\left(3\right)+x_{i}\left(2\right)\right],$$
(4c)$$x_{i+1}\left(3\right)=-x_{i}\left(3\right)e^{-2\beta }+2re^{-\beta }\left[-x_{i}\left(1\right)x_{i}\left(2\right)-x_{i}\left(1\right)x_{i}\left(3\right)+x_{i}\left(3\right)\right],$$
From Equations (4a)–(4c) we can see that x(1) = $\langle \hat{\alpha }\rangle$, $x\left(2\right)=\langle \hat{\alpha }\rangle +\delta \hat{\alpha },$ and $x\left(3\right)=\langle \delta \hat{\alpha }\delta \hat{\alpha }\rangle$ and β is a bifurcation parameter. If $x_{i+1}\left(2\right),\;x_{i+1}\left(3\right)\to 0$ or β→∞ then the system (4) leads to the classical logistic map. When we iterate the system (4) with some specific initial values $x_{0}\left(1\right),\;x_{0}\left(2\right)$ and $x_{0}\left(3\right)$ then we get highly random real values as the output of the chaotic map.

#### 2.2.1. Bifurcation Plots

The effect of dissipation can be observed by plotting a bifurcation diagram. In Figure 1 we have depicted β-bifurcation diagram by fixing the value of *r* as r=3.65, r=3.74 and r=3.90 and ranging β from 2.5 to 6. For small values of the dissipation parameter, the map is on a fixed point, and we get a stable behavior, but with the increase in β we get a period-doubling conversion to chaos. From the bifurcation diagram, we can examine that the strength of quantum correlations is decreased by increasing the value of the dissipation parameter β in the chaotic system and if β approaches to ∞, the system (4) give a classical logistic map. 

#### 2.2.2. Data Randomness Plot

The time series plot can observe randomness in a chaotic map. This graph plots the output of a chaotic map with respect to time. Figure 2 presents first 1000 output values of the quantum chaotic map. Irregularity in data shows that the chaotic quantum system exhibits highly random output.

### 2.3. Fibonacci and Lucas Sequence

Fibonacci numbers, usually represented by $F_{n}$, formed a sequence named Fibonacci sequence and characterized as each number is a sum of two previous numbers, beginning values between 0 and 255, from values 0 and 1 as shown in Figure 3. Mathematically it can be stated as:$$F_{n}=F_{n-1}+F_{n-2},\;\mathrm{with}\;n\geq 2$$
for
$$F_{0}=0,\;F_{1}=1$$

The initial few terms of the Fibonacci sequence are: 0, 1, 1, 2, 3, 5, 8, 13, 21, 34, 55, 89, 144 ….

The Lucas series named after the mathematician Francios Edouard Anatole Lucas is a special case of the Fibonacci sequence that is shown in Figure 4, and defined by the recurrence relation as: $$L_{n}≔\{\begin{array}{c} 2\;if\;n=0;\\ 1\;if\;n=1;\\ L_{n-1}+L_{n-2}\;if\;n\geq 2. \end{array}$$

The first few Lucas terms are 2, 1, 3, 4, 7, 11, 18, 29, 47, 76, 123 ….

## 3. Design of Quantum Image Encryption Scheme

Here, we introduced a new data encryption approach that utilized quantum image method. The system has two layers: a diffusion layer using chaotic quantum map and a confusion layer using an s-box (as substitution operation). A quantum chaotic system is utilized to randomize the input data highly. The application of the s-box increases confusion in cipher image, which breaks the relationship between plain and cipher data. The designed encryption algorithm comprises the two sub-algorithms. Algorithm 1 involves the implementation of a quantum chaotic map on image layers. In Algorithm 2, we have generated S-box using the Lucas series and implemented it on the image layers obtained after the application of the quantum map. The flow chart of the offered encryption technique is depicted in Figure 5.

Further description of algorithm is presented as follows:


**Algorithm 1 Execution of Quantum Chaotic Map**
Firstly, we set diffusion key parameters that are initial conditions and chaotic parameters for quantum chaotic map.1. Select x(1)=0.4634, x(2)=0.0004, x(3)=0.0002 as initial conditions for system (4).2. Set the chaotic parameters as r∈[3.6, 4] and β∈[2.5, 6]. 3. Select image of size m×n and separate layers of the image.4. Iterate chaotic map by using selected initial conditions and chaotic parameters up to
m×n.5. Arrange the output obtained from the chaotic system in an array.6. Sort the image elements according to the array obtained from the chaotic map.7. Combine the data obtained from step 6 and pass it to Algorithm 2.


**Algorithm 2 S-box Generation and Implementation**
The main purpose of Algorithm 2 is to create confusion in data obtained from the algorithm. To add confusion in data we apply the substitution box constructed by using the Lucas series. 1. Use the pseudo-random number to generate random integer arrays. 2. XOR each array with different Lucas series output.3. Select unique 256 (ranging from 0 to 255) elements from array obtained from step 2.4. Compile the elements obtained from step into 16×16 S-box.5. Pass the output obtained from Algorithm 1 through the proposed S-box.6. Compose the resulted data as an encrypted image.

Substitution box generated by proposed method is listed in Table 1.

The proposed substitution box by using the Lucas series is presented in Table 1.

### Experimental Results

This part of the article presents the visual results of the proposed encryption system. We have presented experimental results of Peppers 512 × 512 × 3 image. Figure 6a–d are original layers and Figure 6e–h are their respective cipher layers. We can observe perfect randomness in data visually.

## 4. Design of Proposed Image Decryption Scheme

The decryption procedure is the similar as encryption but in a reversal manner. The input of the decryption algorithm is encrypted image received from the encryption algorithm. The decryption process also comprises of two steps. Firstly, we apply inverse substitution box and then inverse chaotic system. The image decryption process is as follows:

Decryption Algorithm 1: Inverse Substitution Box

In this algorithm, we put the encrypted image as input and apply the inverse substitution box on each layer of image. After the implementation of inverted S-box, the image layers are passed to the decryption algorithm 2.

Decryption Algorithm 2: Inverse Chaotic Map

After the inverse S-box, we apply the inverse chaotic map on the image obtained from decryption algorithm 1. We apply the inverse sorting arrangement of the encryption chaotic map to get the original image.

## 5. Performance Analysis

Mathematical simulations are executed on the MATLAB 2019 platform to validate the efficiency and reliability of the offered quantum-based encryption model. We utilized test color images are Baboon, Peppers, Lena, Fruits, Airplane, House of size 512×512. The robustness of proposed S-box, histogram analysis, correlation coefficient, information entropy, image similarity, randomness and key space analyses are performed in this section. 

### 5.1. Robustness of Proposed S-Box

Multiple S-box tests are applied to analyze the reliability as well as the robustness of the designed S-box. The designed S-box is used in our encryption model. Four necessary S-box tests are applied to designed S-box to check effectiveness and validity. The assessments here include non-linearity test analysis (NL), strict avalanche criterion (SAC), bit Independence criterion (BIC), and differential approximation probability (DP). The result obtained using the proposed S-box is compared to the existing S-boxes. Each test result depicts that constructed S-box for encryption model has a better ability to resist any attack. 

Non-linearity is an essential criterion for finding the strength of encrypted information by the process of substitution. From Ref. [41] non-linearity (Ng) is elaborated in more detailed. The value Ng should be higher for the validation of the robustness of the model. The minimum calculated value of Ng for the designed S-box is 104 while the maximum computed value is 108, with an average value is approximately 105.225. As shown in Table 2, the designed S-box value is higher compared to existing S-box non-linearity values. 

Strict avalanche criterion (SAC) is a ratio of change in bits to the number of bits in the ciphertext. The test is more detailed in Ref. [41]. Depicting from Table 3 values of SAC is spread over the maximum to a minimum. The optimal value of SAC is 0.5. The values are near to 0.5, as shown in Table 3 and Table 4, which shows that the designed S-box is robust for any linear and differential attacks. 

Bit independent criterion non-linearity (BIC-NL) and bit independent criterion strict avalanche criterion (BIC-SAC) is used to find the strength of designed S-box. The computed values must be higher to existing S-boxes BIC-NL and BIC-sac values. The test is more detailed in Ref. [41]. Results and comparison of both tests are shown in Table 5 and Table 6, respectively. The achieved results are quite exceptional as compared to presented S-box results.

The differential approximation is another essential criterion that is widely used to find the reliability of S-box. The lower the value of $DP_{g}$ indicates greater resistivity to differential attacks. The computed value of 12 (see Table 7) shows that the S-box designed for encryption model is highly secure. The test is more detailed in [41].
(5a)$$\mathrm{Nonlinearity}\;\mathrm{test}\;(\mathrm{NL})=N_{g}=2^{k-1}(1-2^{k}\max_{\varphi \in GF(2^{k})}|S_{\left(g\right)}(\varphi )|),$$
(5b)$$S_{\left(g\right)}(\varphi )=\sum_{\varphi \in GF(2^{k})}{\left(-1\right)}^{x.\varphi ⊕g(x)},$$
(6a)$$\mathrm{Strict}\;\mathrm{avalanche}\;\mathrm{criterion}=S(g)=\frac{1}{k^{2}}\sum_{1<r\leq k}\sum_{1\leq \omega \leq k}\left|\frac{1}{2}-Q_{r,\omega }(g)\right|,$$
(6b)$$Q_{r,\omega }(g)=2^{-k}\sum_{x\in B^{k}}g_{\omega }(x)⊕g_{\omega }(x⊕e_{r}),$$
where $e^{r}={\left[\theta_{r,1}\theta_{r,2}\ldots \theta_{r,k}\right]}^{T}$ and
$$\theta_{r,\omega }=\{0,r\neq \omega \}$$
$$\theta_{r,\omega }=\{1,r=\omega \}$$
(7)$$DP_{g(\partial k\to \partial l)}=\left[\frac{\#\{k\in \frac{X}{S(k)}⊕S\left(k⊕\partial k\right)=\partial l\}}{2^{m}}\right]$$

### 5.2. The Histogram Analysis

It is one of the crucial measures to evaluate the working of encryption technique. The analysis reveals the pixels frequency distribution of an image. An ideal encryption algorithm always produces ciphers which create a uniform histogram for any original data. Figure 7 presents simulation results of the histogram for original and encrypted image layers of Peppers. It can be noticed that the histogram of the encipher images is drastically dissimilar from the original ones. According to the depicted results in Figure 7, the offered encryption scheme is perfect for opposing all histogram related attacks.

### 5.3. Correlation Coefficient Analysis 

Pixels of original images having significant visual content are highly correlated to each other in horizontal, diagonal and, vertical way. Correlation coefficient value is 1 for original images. A perfect encryption algorithm must reduce this correlation value in each direction. Therefore, it is evident that after the implementation of the robust encryption scheme, the correlation reduces to 0. We have performed the correlation analysis for some standard images. We have chosen 10,000 pairs of pixels from initial and encipher images, and we have computed the correlation coefficient among neighboring pixel values as follows:(8)$$\gamma_{xy}=\frac{\left|\frac{1}{M}\sum_{i=1}^{M}\left(x_{i}-mean\left(x\right)\right)\left(y_{i}-mean\left(y\right)\right)\right|}{\sqrt{\frac{1}{M}\sum_{i=1}^{M}{\left(x_{i}-mean\left(x\right)\right)}^{2}}\sqrt{\frac{1}{M}\sum_{i=1}^{M}{\left(x_{i}-mean\left(x\right)\right)}^{2}}},$$
where *x* and *y* are grey-level pixels of two neighboring pixels, *M* is the overall number of adjacent pixels in original and cipher image. Results of correlation analysis are listed in Table 8, also we have performed some comparative analysis with some existing algorithms. 

Depicted results in Table 8 claims that the neighboring pixels of original and cipher images are uncorrelated by the implementation of the suggested scheme as comparative to other existing algorithms.

The visual analysis of correlation can be performed by marking the distribution of neighboring pixels of original and its respective cipher image using graph. The correlation distribution of each direction is given in Figure 8, Figure 9 and Figure 10 for different layers of Peppers image. From the Figures, it can be visualized that there is a clear alteration among the plain and enciphered correlation diagram, which shows the robustness of the offered encryption model.

### 5.4. Information Entropy

Randomness based on information entropy is one of the crucial properties to evaluate the uncertainty of data. Entropy can be computed by:(9)$$H\left(x\right)=-\overset{2^{n}-1}{\underset{i=0}{\sum }}P\left(x_{i}\right)\log_{2}P\left(x_{i}\right),$$
where $P\left(x_{i}\right)$ is the probability distribution of each *x*. The ideal value of entropy for cipher image with 256 gray level is 8. Therefore, an encryption scheme must be idyllic if it yields cipher with entropy value close to 8. The entropy results for some standard images are presented in Table 9. Stated calculations in Table 9 shows that entropy of our proposed encryption method is almost close to 8, which indicates that the output data is highly randomized. 

Moreover, we have presented a brief comparison offered scheme with some existing work in the literature (see Table 10). Comparative outcomes reveal the overall performance of the offered encryption method.

### 5.5. Plaintext Sensitivity Analysis

When we alter one pixel of the original image, then ciphertext must be 50% changed to ensure the offered scheme’s privacy. To observe the plaintext sensitivity in the suggested technique, we have calculated the number of pixels changing rate (NPCR) and the unified average changing intensity (UACI). These analyses are performed on two images, the first one is the enciphered image of the original image, and the second one is the cipher image of one-pixel change original image. Mathematically NPCR and UACI are demonstrated as:(10)$$\mathrm{NPCR}=\frac{\sum_{i=1}^{M}\sum_{j=1}^{N}D\left(i,j\right)}{M\times N}\times 100,$$
(11)$$D\left(i,j\right)=\{\begin{array}{c} 1,\;C\left(i,j\right)\neq C^{\prime }\left(i,j\right),\\ 0,\;C\left(i,j\right)=C^{\prime }\left(i,j\right),\; \end{array}$$
(12)$$\mathrm{UACI}=\frac{1}{M\times N}\left(\overset{M}{\underset{i=1}{\sum }}\overset{N}{\underset{j=1}{\sum }}\frac{\left|C\left(i,j\right)-C^{\prime }\left(i,j\right)\right|}{255}\right)\times 100,$$
where M×N is the size of cipher images. We have listed calculated test results of NPCR and UACI of some standard images in Table 11. The standard result values of NPCR and UACI statistical test are 99.61 and 33.44 for a secure encryption scheme. The average values of NPCR and UACI of the suggested scheme are much better than existing schemes. Comparative results indicate that our offered scheme is robust against chosen-plaintext attack.

### 5.6. Image Quality Measures

In the progress of image processing algorithms, image quality measurement (IQM) plays a vital role. An extensive set of analyses are conducted to ensure the quality of the encrypted image. The mathematical formulation for quality evaluations are as follows:(13)$$\mathrm{MSE}=\frac{1}{M\times N}\overset{\;}{\underset{i,j}{\sum }}{\left(P\left(i,j\right)-P^{\prime }\left(i,j\right)\right)}^{2},$$
(14)$$\mathrm{PSNR}=10\log\frac{{\left(2^{n}-1\right)}^{2}}{MSE},$$
(15)$$\mathrm{NCC}=\frac{\sum_{i,j}^{\;}P\left(i,j\right)\to P^{\prime }\left(i,j\right)}{\sum_{i,j}^{\;}P{\left(i,j\right)}^{2}},$$
(16)$$\mathrm{AD}=\frac{\sum_{i,j}^{\;}\left(P\left(i,j\right)-P^{\prime }\left(i,j\right)\right)}{M\times N},$$
(17)$$\mathrm{SC}=\frac{\sum_{i,j}^{\;}P{\left(i,j\right)}^{2}}{\sum_{i,j}^{\;}P^{\prime }{\left(i,j\right)}^{2}},$$
(18)$$\mathrm{MD}=\mathrm{Max}\left(P\left(i,j\right)-P^{\prime }\left(i,j\right)\right),$$
(19)$$\mathrm{NAE}=\frac{\sum_{i,j}^{\;}{\left(P\left(i,j\right)-P^{\prime }\left(i,j\right)\right)}^{2}}{\sum_{i,j}^{\;}\left|P\left(i,j\right)\right|},$$
where M×N is the total dimension of plain and cipher image, P(i,j) denotes the plain image and $P^{\prime }\left(i,j\right)$ represents its corresponding cipher image. Image quality measures for some standard color images are depicted in Table 12. The proposed scheme image quality values are furthered compared to Younas et al. [51] mean values taken from each layer as shown in Table 12. 

### 5.7. Key Space Assessment

The private key applied to encrypt the image data would neither be too wide nor too small. Too wide private key limits the pace of encryption and is not suitable for real-time data communication, whereas a shorter private key lead in a brute force attack. The length of the key space must not be less than $2^{100}$ to include a high degree of protection from the perspective of cryptography. In the proposed scheme we are using five private keys in Quantum map from which three are initial conditions and the other two are chaotic parameters. Each key provides the accuracy level of $2^{25}$ and hence the key space for this algorithm is $2^{125}$. Moreover, the substitution box generated from the Lucas series provides confusion in the ciphertext. Our offered encryption scheme presents an idea with a greater key space which enhances the convolution of the system. In addition, we have presented key sensitivity analysis in Table 13.

### 5.8. Time Complexity Analysis

Minimum computational cost and resources should be used for an efficient encryption algorithm. We have computed the time taken to encrypt each image by using MATLAB 19. We have also listed some comparative analyses in Table 13. As decryption of the algorithm is the reverse of encryption therefore time utilized for decryption is the same. In comparison with existing schemes, it is seen from Table 14 that the proposed scheme has less computational complexity.

### 5.9. Randomness Test

The security of encryption algorithm can be guaranteed by some features as efficiency, fair distribution, and complexity. To check all these properties, we have performed NIST SP800-22 test suit [54] on the encipher image generated by using the suggested encryption technique. It is by far, one of the most comprehensive assessment criteria. It is conventional that NIST SP 800-22 assessments are used for 0–1 sequences so that the cipher image can be viewed as a binary data stream format. In Table 15 we have listed the NIST results for Peppers image layers. 

## 6. Conclusions

An innovative image encryption model focused on a chaotic quantum map is described in this work. Before the implementation of the chaotic map, a new design of substitution-box technique based on the Lucas sequence is offered. The mixture of the chaotic quantum map and the substitution-box provides perfect security level for data transmission. The proposed scheme is passed through some standard security performance assessment to examine the competence of the offered encryption technique. The experiments result, and statistical analyzes demonstrate that the offered method has increased reliability and effectiveness toward multiple statistical and differential attacks.

## Figures and Tables

**Figure 1 entropy-22-01276-f001:**
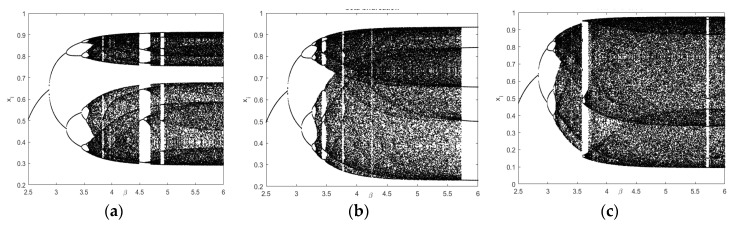
Adjacent to *x*-direction at (**a**) r=3.65; (**b**) r=3.74; (**c**) r=3.90.

**Figure 2 entropy-22-01276-f002:**
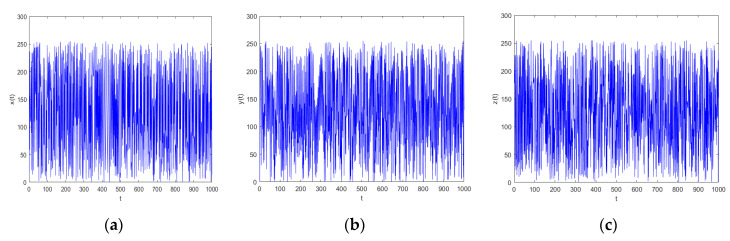
Quantum chaotic map output plot along (**a**) *x* and *t*; (**b**) *y* and *t*; (**c**) z and *t*.

**Figure 3 entropy-22-01276-f003:**
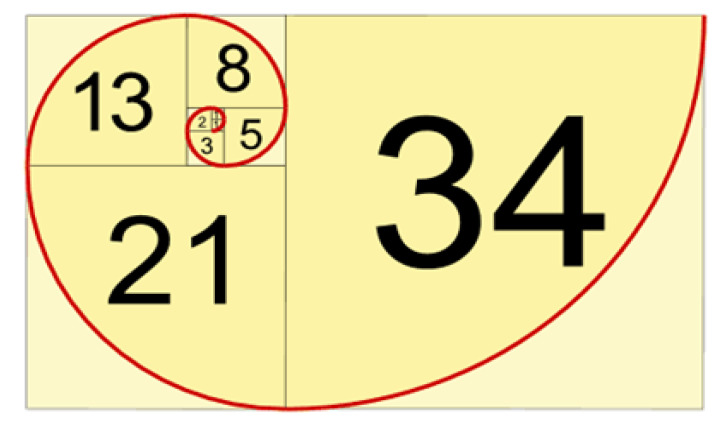
The Fibonacci spiral [39].

**Figure 4 entropy-22-01276-f004:**
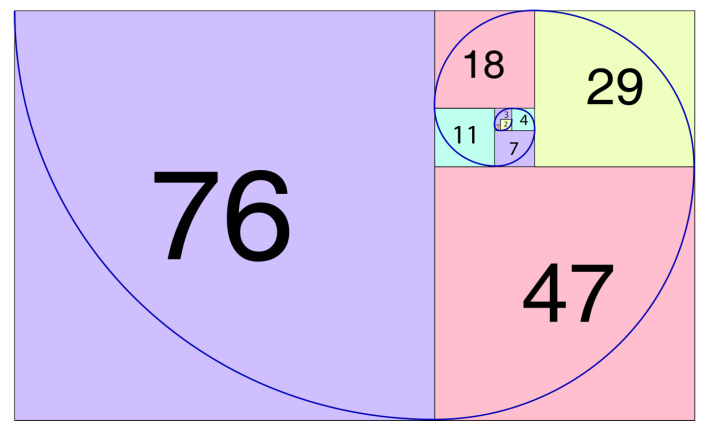
The Lucas spiral [40].

**Figure 5 entropy-22-01276-f005:**
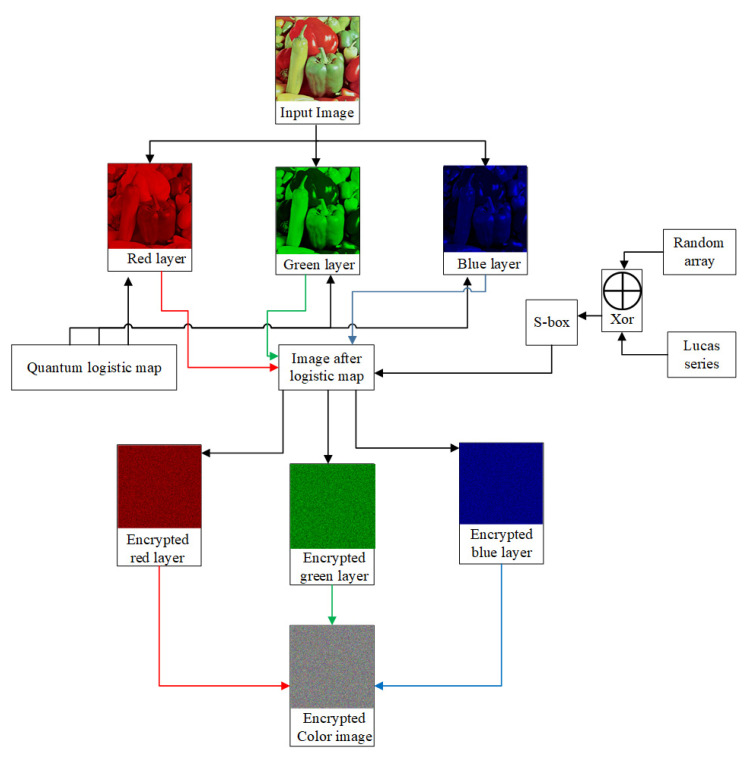
Design of offered encryption technique.

**Figure 6 entropy-22-01276-f006:**
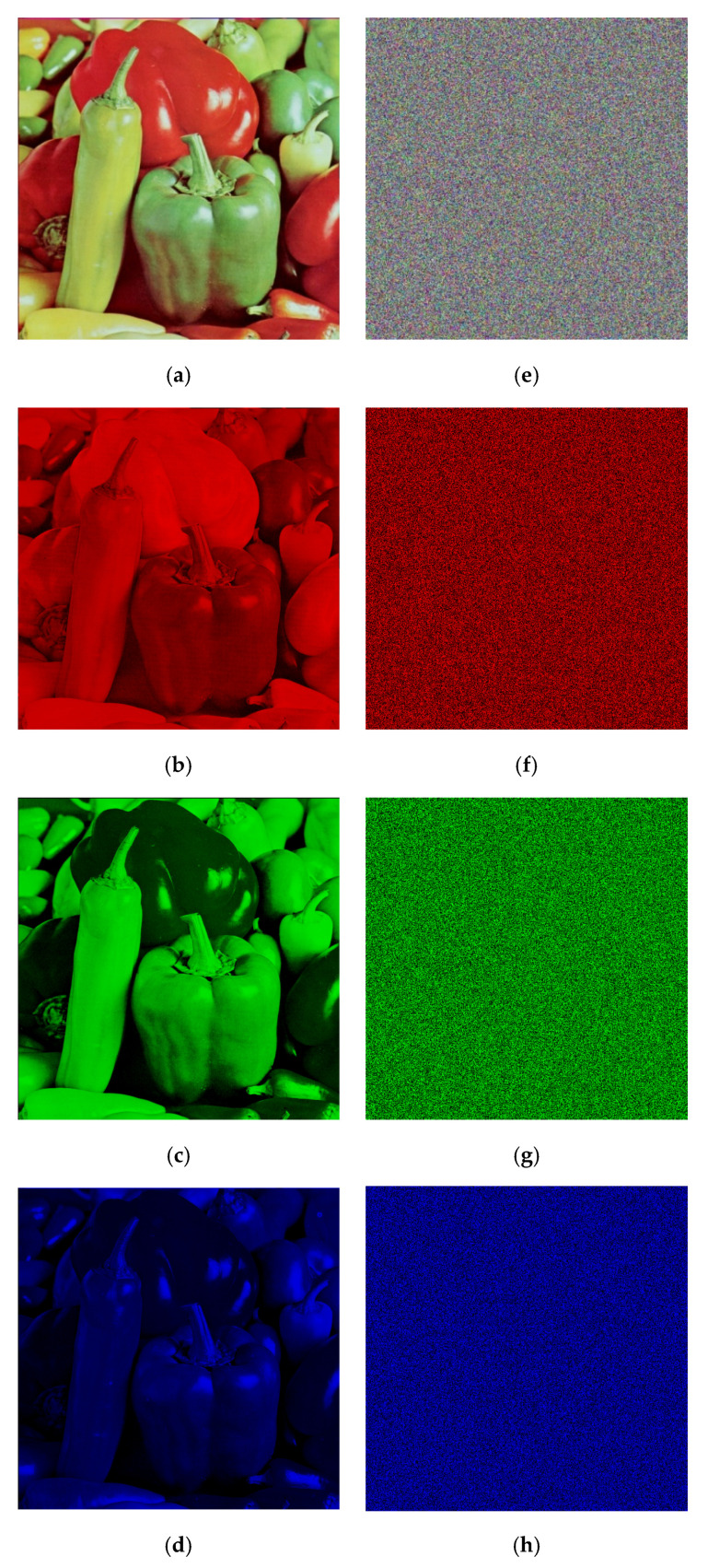
Experimental results of proposed algorithm on Peppers image. (**a**) Plain colored image; (**b**) Plain red channel; (**c**) Plain green channel; (**d**) Plain blue channel; (**e**) Encrypted colored image; (**f**) Encrypted red channel; (**g**) Encrypted green channel; (**h**) Encrypted blue channel.

**Figure 7 entropy-22-01276-f007:**
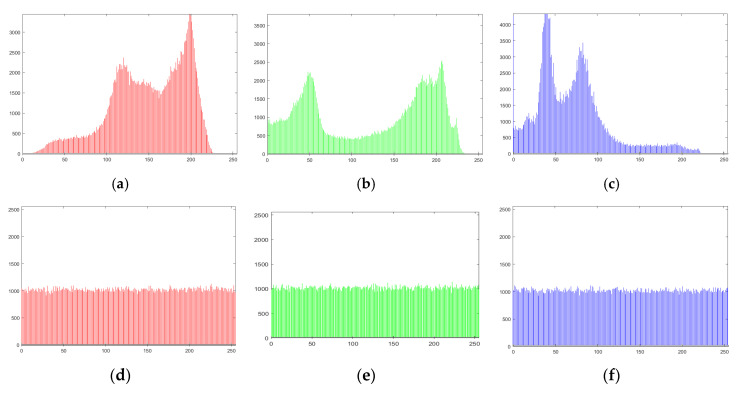
Peppers image histogram (**a**–**c**) Original layers; (**d**–**f**) Encrypted layers.

**Figure 8 entropy-22-01276-f008:**
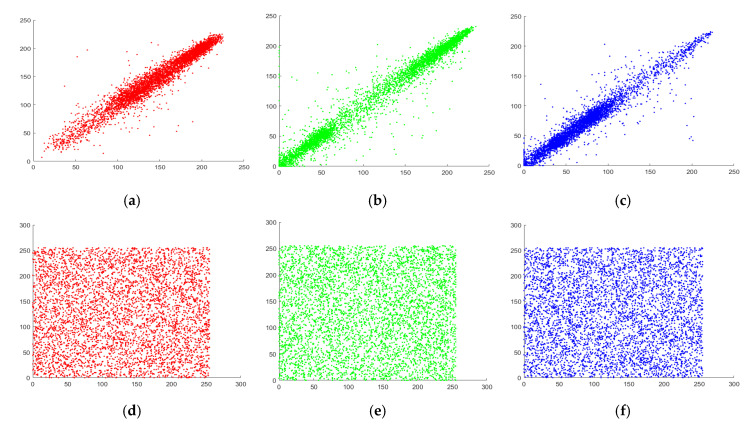
Correlation diagram in horizontal direction for Peppers image (**a**–**c**) Original channels; (**d**–**f**) Encrypted channels.

**Figure 9 entropy-22-01276-f009:**
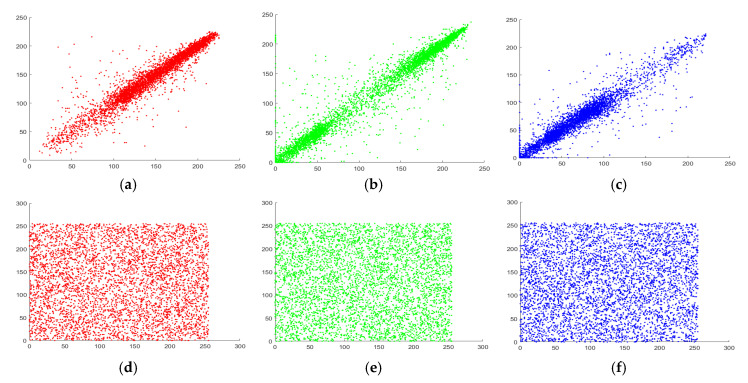
Correlation diagram in diagonal direction for Peppers image (**a**–**c**) Original channels; (**d**–**f**) Encrypted channels.

**Figure 10 entropy-22-01276-f010:**
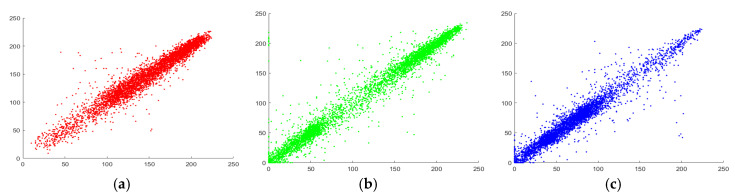
Correlation diagram in vertical direction for Peppers image (**a**–**c**) Original channels; (**d**–**f**) Encrypted channels.

**Table 1 entropy-22-01276-t001:** Proposed substitution box.

242	189	222	111	3	100	253	206	78	24	15	226	219	88	93	137
215	37	194	79	114	87	84	156	254	163	134	234	58	245	155	169
120	43	132	220	136	62	145	17	230	191	21	33	209	77	67	178
68	36	32	249	29	246	188	117	142	85	202	240	57	25	0	89
251	97	95	71	41	200	247	227	54	60	133	203	161	146	182	45
198	231	116	1	125	158	72	42	40	5	208	103	180	190	236	210
218	170	147	47	129	192	207	46	76	73	153	52	106	59	99	185
20	11	159	107	187	80	123	10	151	101	13	165	105	141	48	233
91	232	8	183	109	239	7	118	157	6	138	44	238	56	213	104
96	121	199	171	144	23	241	201	179	154	205	214	94	9	162	135
244	38	63	143	2	108	166	167	212	75	14	98	30	110	216	148
174	186	16	90	217	221	150	49	181	235	255	83	228	119	177	4
172	252	224	164	53	61	175	74	26	51	82	39	193	86	19	126
140	195	197	225	237	248	35	211	92	65	112	223	31	69	131	55
196	173	127	70	152	124	66	184	128	122	34	64	28	115	50	22
18	204	139	130	250	176	168	12	102	229	149	243	27	160	81	113

**Table 2 entropy-22-01276-t002:** Nonlinearity test for constructed S-box and comparison.

S-Box	Max	Min	Mean
Constructed S-box	108	104	105.25
Existing S-box [42]	108	100	103.25
Existing S-box [43]	109	103	104.88
Existing S-box [44]	106	100	103
Existing S-box [45]	106	100	103.25
Existing S-box [46]	108	102	104.75
Existing S-box [47]	108	98	103

**Table 3 entropy-22-01276-t003:** SAC dependence matrix of designed S-box.

0.5469	0.5000	0.5000	0.5000	0.5000	0.5000	0.5156	0.5469
0.4531	0.5391	0.5234	0.4531	0.5000	0.5000	0.5156	0.5000
0.5469	0.4609	0.5234	0.4531	0.5313	0.4844	0.5156	0.5469
0.5000	0.4609	0.5000	0.5000	0.5313	0.5156	0.5000	0.4531
0.5000	0.5000	0.4766	0.4531	0.5000	0.4844	0.4844	0.5469
0.5000	0.4609	0.4766	0.4531	0.5313	0.5156	0.5156	0.5000
0.5000	0.5000	0.5000	0.5000	0.4688	0.5156	0.4844	0.4531
0.5000	0.4609	0.5000	0.5000	0.5000	0.5000	0.5156	0.5000

**Table 4 entropy-22-01276-t004:** Analysis of strict avalanche criterions (SAC) for different S-boxes.

S-Boxes	Max	Min	Mean
The obtained S-box	0.5469	0.4531	0.4987
Existing S-box [42]	0.5938	0.3750	0.5059
Existing S-box [43]	0.5703	0.3984	0.4966
Existing S-box [44]	0.6094	0.4219	0.5000
Existing S-box [45]	0.5938	0.4219	0.5049
Existing S-box [46]	0.5938	0.3906	0.5056
Existing S-box [47]	0.5938	0.4063	0.5012

**Table 5 entropy-22-01276-t005:** Bit independent criterion (BIC)-Nonlinearity (NL) for designed S-box.

0	118	116	108	110	116	114	114
118	0	116	110	106	110	114	104
116	116	0	108	116	110	116	114
108	110	108	0	110	114	118	116
110	106	116	110	0	118	118	110
116	110	110	114	118	0	108	108
114	114	116	118	118	108	0	114
114	104	114	116	110	108	114	0

**Table 6 entropy-22-01276-t006:** Comparison of Bit independent criterion (BIC)-nonlinearity (NL) with different S-boxes.

S-Box	BIC-SAC	BIC-Nonlinearity
Constructed S-box	0.4990	112.64
Existing S-box [42]	0.5031	104.29
Existing S-box [43]	0.5044	102.96
Existing S-box [44]	0.5024	103.14
Existing S-box [45]	0.5010	103.71
Existing S-box [46]	0.5022	104.07
Existing S-box [47]	0.4989	104.07

**Table 7 entropy-22-01276-t007:** Differential approximation for constructed S-box.

6	6	8	6	8	8	8	6	6	6	6	8	6	12	8	6
8	8	6	6	6	6	6	6	6	8	8	6	6	8	6	6
8	6	6	8	8	10	8	6	8	6	6	6	8	6	8	10
6	6	8	4	6	6	6	6	6	6	6	6	8	6	6	6
6	8	4	6	6	6	6	6	6	6	8	6	6	10	8	6
8	8	6	8	6	6	8	8	6	6	6	6	8	8	6	6
6	6	6	8	6	6	6	8	6	6	8	8	6	8	6	8
8	6	6	8	6	6	6	6	6	8	6	6	6	6	8	4
10	4	6	6	6	6	8	6	8	6	6	6	6	6	8	6
6	6	6	8	6	4	8	6	6	6	6	8	6	6	8	8
6	8	8	6	6	6	6	8	6	6	6	6	8	6	6	8
6	6	8	6	6	6	8	8	6	8	8	6	6	6	8	6
6	10	8	6	6	6	6	6	6	8	6	8	6	8	6	6
6	6	6	6	6	8	6	8	6	6	6	6	8	6	6	6
6	6	10	8	6	6	6	6	6	6	6	6	6	8	6	6
8	8	8	6	6	6	6	6	8	10	6	8	6	6	6	-

**Table 8 entropy-22-01276-t008:** Correlation coefficient for proposed scheme and some existing algorithms.

		Proposed Scheme		Ref. [48]	Ref. [49]
Image	Direction	Plain Image	Cipher Image		
Baboon	Horizontal	0.9231	0.0009	0.003984	−0.0038
	Diagonal	0.8543	0.0013	0.003949	0.0003
	Vertical	0.8660	−0.0016	−0.004631	0.0007
Peppers	Horizontal	0.9635	0.0013	−0.000116	−0.0009
	Diagonal	0.9564	0.0007	−0.002276	0.0033
	Vertical	0.9663	−0.0015	−0.000307	0.0008
Airplane	Horizontal	0.9726	0.0016	−0.001662	0.0006
	Diagonal	0.9343	−0.0030	0.003358	−0.0011
	Vertical	0.9568	−0.0008	0.000894	0.0029
House	Horizontal	0.9671	−0.0010	−0.002882	-
	Diagonal	0.9126	0.0005	0.004594	-
	Vertical	0.9353	0.0002	−0.004121	-

**Table 9 entropy-22-01276-t009:** Information entropy results for some standard images.

Image Name	Plain Image			Encipher Image		
	R	G	B	R	G	B
Baboon	7.7067	7.4744	7.7522	7.9992	7.9994	7.993
Lena	7.5889	7.1060	6.8147	7.9993	7.9993	7.9993
Peppers	7.3388	7.4963	7.0583	7.9992	7.9993	7.9992
Fruits	7.0556	7.3527	7.7134	7.9993	7.9993	7.9993
Airplane	6.7178	6.7990	6.2138	7.9993	7.9993	79993
House	7.4156	7.2295	7.4354	7.9993	7.9993	7.9994

**Table 10 entropy-22-01276-t010:** Comparative information entropy analysis.

Image Name	Suggested Scheme			Ref. [48]		
	R	G	B	R	G	B
Baboon	7.9992	7.9994	7.993	7.99930	7.99934	7.99929
Peppers	7.9992	7.9993	7.9992	7.99923	7.99922	7.99937
Airplane	7.9993	7.9993	79993	7.99930	7.99937	7.99931
House	7.9993	7.9993	7.9994	7.99932	7.99932	7.99937

**Table 11 entropy-22-01276-t011:** NPCR and UACI results for offered scheme and comparison with existing technique.

	Offered Scheme		Ref. [50]	
Image	NPCR	UACI	NPCR	UACI
Baboon	99.60	33.49	99.12	33.11
Lena	99.61	33.51	99.22	33.12
Peppers	99.61	33.50	99.15	33.14
Airplane	99.61	33.48	99.18	33.11
House	99.61	33.48	98.87	32.16

**Table 12 entropy-22-01276-t012:** Image quality measures of proposed scheme for some standard images.

	MSE	PSNR	NCC	AD	SC	MD	NAE
Baboon	0.00042	11.8481	0.8878	2.2558	0.9958	203	0.4084
Lena	0.00053	10.8339	1.0999	−31.7019	0.5938	200	0.6232
Peppers	0.00054	10.7852	0.8839	−7.4623	0.9256	211	0.4970
Fruits	0.00063	10.1128	0.7082	37.3536	1.5872	244	0.3951
Airplane	0.00072	9.5193	0.6670	51.6376	1.8315	217	0.3984
House	0.00058	10.4310	0.7263	33.8372	1.5148	223	0.3898
Comparison of image quality measure with the average calculated values of Younas et al. [51]							
Lena	-	8.6290	0.9145	2.3396	0.8333	242	0.6456
Baboon	-	8.7486	0.8813	−8.6360	0.7333	243	0.6527
Airplane	-	7.9353	0.6614	54.2964	1.5666	249	0.4600
Pepper	-	8.1351	0.9639	−16.6711	0.7566	243	0.8420

**Table 13 entropy-22-01276-t013:** Key sensitivity test.

Original Encryption Key	Wrong Decryption Key	Decryption
x(1)=0.4634, x(2)=0.0004, x(3)=0.0002, r=3.74, β=2.6	x(1)=1.4634, x(2)=0.0004, x(3)=0.0002, r=3.74, β=2.6	Fail
	x(1)=0.4634, x(2)=0.0004, x(3)=0.0002, r=3.74, β=2.9	Fail
	x(1)=0.4634, x(2)=0.0004, x(3)=0.0002, r=3.84, β=2.6	Fail

**Table 14 entropy-22-01276-t014:** Time (Sec) taken for encryption.

Image	Proposed	Ref. [52]	Ref. [53]
Baboon	1.06	3.53	11.45
Lena	1.13	3.23	11.12
Peppers	1.02	3.68	12.13
Fruits	1.26	-	-
Airplane	1.25	-	-
House	1.09	-	-

**Table 15 entropy-22-01276-t015:** NIST SP800-22 test suit results for proposed encryption scheme.

Test Name	*p* Values		
	Red Layer	Green Layer	Blue Layer
Frequency	0.59086	0.22949	0.92442
Block-frequency	0.847	0.57168	0.43369
Runs (*M* = 10,000)	0.61607	0.1394	0.97489
Long runs of ones	0.035752	0.035752	0.035752
Rank	0.29191	0.29191	0.29191
DFT	0.38399	0.66336	0.99881
No overlapping templates	098566	0.98566	0.98566
Overlapping templates	0.85988	0.85988	0.85988
Universal	0.9931	0.99908	0.99659
Approximate entropy	0.95452	0.72847	0.93672
Cumulative sums (1)	0.24299	0.23783	0.27354
Cumulative sums (2)	0.94041	0.5349	0.85465

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
