# Peer review of "A Digital Image Confidentiality Scheme Based on Pseudo-Quantum Chaos and Lucas Sequence"

_entropy, 2020, doi:10.3390/e22111276_

Round 1
Reviewer 1 Report
The contribution of the proposed work is not sufficient and the paper suffers from precision. The authors claim perfect security and good computing performance without any proof. So, the manuscript cannot be accepted for publication in Entropy.
Critical Comments:
- As the main contribution of the paper is the design of a new s-box based on Lucas sequence. Then, it is necessary, first to justify the use of Lucas sequence and especially to verify the robustness of the proposed s-box by applying some tests: bijectivity, non linearity, strict avalanche criterion (SAC), output bits independence criterion (BIC), and equiprobable input/output XOR distribution. After that you can compare the performance of the proposed s-box with the performance of other published S-boxes.
- The computational performance of the proposed encryption system, as the encryption throughput (ET) and the needed number of cycles to encrypt one Byte (NCpB), is not at all studied. The last parameter allows you to compare the complexity of cryptosystems regardless of the machine used.
Some observation and comments:
Abstact:
Line 21, replace efficiency by efficiency in terms of statistical analysis.
Introduction:
Line 63: replace diffusion by confusion.
Paragraph 2.1.2:
You must mention an operation of quantization allowing to have values between 0 and 255, from values 0 and 1.
Section 3.
Line 142, replace:
This approach focused on two concepts, chaotic quantum map, and substitution box.
By
The system has two layers: a diffusion layer using chaotic quantum map and a confusion layer using an s-box (as substitution operation).
Section 4:
It is necessary to detail how you build the inverse s-box.
You can’t obtain an inverse chaotic system for your chaotic syste (in the decryption side, you use the same chaotic system in your cryptosystem)
Section 5:
In formula (5), replace logarithm in base e by logarithm in base 2
Etc…
Author Response
Answers to Remarks, Comments, and Suggestions -- Reviewer 1
General:
We are extremely thankful to the anonymous reviewer for the useful comments and suggestions that have helped us to improve the quality of this manuscript. This document contains answers to the questions and comments by the reviewer. The file contains answers to questions by the reviewer in the section, ‘Comments and Suggestions for Authors’ of the review report.
Answers to Comments from Reviewer 1
Comment 1:
As the main contribution of the paper is the design of a new s-box based on Lucas sequence. Then, it is necessary, first to justify the use of Lucas sequence and specially to verify the robustness of the proposed s-box by applying some tests: bijectivity, non-linearity, strict avalanche criterion (SAC), output bits’ independence criterion (BIC), and equiprobable input/output XOR distribution. After that you can compare the performance of the proposed s-box with the performance of other published S-boxes.
The computational performance of the proposed encryption system, as the encryption throughput (ET) and the needed number of cycles to encrypt one Byte (NCpB), is not at all studied. The last parameter allows you to compare the complexity of cryptosystems regardless of the machine used.
Answer to Comment 1:
Thank you so much for your comments. The comments really helped me enhancing the quality of the work. The work has been substantially enhanced in the modified manuscript as per your valuable suggestions. Respected reviewer, we have designed/presented an encryption algorithm in which S-box is used just to add confusion to the system (As a part). The main purpose/intension of our scheme is to reduce the cost of the proposed scheme and computational time by developing an innovative schemes. The encryption algorithm (proposed scheme) is validated using a number of statistical analyses. Here we do not need to analyze S-box independently. However, we modified some of the tests, i.e., Key Space Assessment (Line 334-344). The results are added with remarks that the decryption has failed. Please check the line (348-354) for Time complexity. The system takes very little time for execution and encryption. Please check Table 7 and 8. The results are compared to various existing schemes. Please see Ref [39] and Ref [40] as shown in Table 8. Depicted from the computational speed and existing scheme time complexity result. The proposed scheme result is much better, i.e., around 1 second that is too less and will take low cost as well.
Section wise:
Comment 2:
Line 21 replace efficiency by efficiency in terms of statistical analysis.
Answer to Comment 2:
Thank you so much for your comments. The comments really helped me enhancing the quality of the work. The work is substantially enhanced as per your valuable suggestions in the modified version of the manuscript. Respected reviewer, I corrected the sentence. Please see Line 21.
- A. Line 63: replace diffusion by confusion.
Answer: The work is thoroughly checked for grammatical mistakes and other technical issues. We changed the word diffusion with confusion. Please see line 65 in the newly modified manuscript. Other grammatical mistakes and other technical issues are also corrected. Thank you so much.
- B. You must mention an operation of quantization allowing to have values between 0 and 255, from values 0 and 1.
Answer: Thank you so much I changed the work accordingly as per your suggestions. Please see line 144 to 145.
- C. Line 142, replace:
This approach focused on two concepts, chaotic quantum map, and substitution box.
By
The system has two layers: a diffusion layer using a chaotic quantum map and a confusion layer using an s-box (as substitution operation).
Answer: The work is modified as per your suggestions. Please see line 167 to 169.
- D. It is necessary to detail how you build the inverse s-box.
You cannot obtain an inverse chaotic system for your chaotic system (in the decryption side, you use the same chaotic system in your cryptosystem).
Answer: Thank you so much for your valuable time. The question is interesting. We can invert our S-box by replacing input with output.
- In formula (5), replace logarithm in base e by logarithm in base 2
Answer: Thank you so much. The work is substantially changed apart from your comments. I replaced logarithm in base e by logarithm in base 2. Please see Line 280.
Apart from your suggestion. I changed some of the other parts in the original draft. Addition of classical chaotic map. The introduction is also changed. Addition of tests. Added references from 30 to 40 relevant to work.

Reviewer 2 Report
1. Please improve the introduction of the paper by describing the logistic classical map.
2. Please, use a large database in your test
3. Table 2: please compare with other references (You just compared with only one reference)
4. Add references of papers that use the logistic map in several encryption algorithms:
https://doi.org/10.1016/j.cnsns.2010.09.005
Author Response
Answers to Remarks, Comments, and Suggestions -- Reviewer 2
General:
We are extremely thankful to the anonymous reviewer for the useful comments and suggestions that have helped us to improve the quality of this manuscript. This document contains answers to the questions and comments by the reviewer. The file contains answers to questions by the reviewer in the section, ‘Comments and Suggestions for Authors’ of the review report.
Answers to Comments from Reviewer 2
Comment 1:
Please improve the introduction of the paper by describing the logistic classical map.
Answer to Comment 1:
Thank you so much for your comments. The comments really helped me enhancing the quality of the work. The work has been substantially enhanced in the modified manuscript as per your valuable suggestions. Respected reviewer, please see section 2.1 (Line 74 to 89). I have added a classical logistic map as per your valuable suggestion. The classical map will surely help the reader to know the basics of chaotic maps prior to the Quantum chaotic systems. Thank you so much once again for valuable comments that enhanced the quality of the work.
Comment 2:
Please, use a large database in your test.
Answer to Comment 2:
Thank you so much for the valuable comment. Please see Table 2, 3, 4, 5, 6, 8. We utilized six different images for the encryption process. Moreover, the work is substantially changed as per your comments. Each modified part is red highlighted in my modified manuscript.
Comment 3:
Table 2: please compare with other references (You just compared with only one reference)
Answer to Comment 3: Table 2 is modified, and additional reference is added for better comparison as per your valuable suggestions. The work is compared to three different images i.e., Baboon, Pepper, and Airplane having the same size.
Comment 4:
Add references to papers that use the logistic map in several encryption algorithms:
https://doi.org/10.1016/j.cnsns.2010.09.005
Answer to Comment 4:
Thank you so much for your comments. The comments really helped me enhancing the quality of the work. The work has been substantially enhanced in the modified manuscript as per your valuable suggestions. We added a number of references relevant to work. Around 12 references are added to the modified manuscript. Please see the reference 28 to 40.

Reviewer 3 Report
1. Recommendation Reconsider after major revision 2. Overview and general recommendation The manuscript deals with quantum colour image encryption method focused on the Lucas series-based substitution box to enhance the competence of encryption. One of the key aspects of the paper is “To reduce cost and time by means of new approach to quantum image encryption.” The paper presents some potential, but the authors should provide some amendments and most of all, they should stress what is the novelty of the paper compared with other related manuscripts. 2.1. Major comments: a) Fig. 1: Provide some test for r in values where the logistic maps present non chaos (around 3.82, for instance). b) line 160: Chaotic parameters: define the reliable values for r and beta. c) line 172: step 3 should be clarified. How is it done? d) How is Table 1 defined? e) Image quality measures should be compared or given some analysis to check if the values are good or not. f) line 311: there is no estimation on the key space. The authors present only a general perspective and qualitative information. g) Some computational cost should be provided, as one of the key features of the paper is to reduce the cost. h) The authors should also present some tests to lost of information. Please refer to the following reference: “Image encryption based on the pseudo-orbits from 1D chaotic map,” Chaos: An Interdisciplinary Journal of Nonlinear Science. Available: http://aip.scitation.org/doi/10.1063/1.5099261 i) Please, stress what is the novelty of the paper compared with other related manuscripts. 2.2 Minor comments: a) l. 28: Revise the sentence: “A wide range of physical happenings can be prescribed by utilizing mathematical models.” b) It would also improve the paper if the figure captions would be made more self-contained. In addition to what is shown for which parameter values, one could also consider a sentence or two saying what is the main message of each figure.Author Response
Answers to Remarks, Comments and Suggestions –Reviewer 3
General:
We are extremely thankful to the anonymous reviewer for the useful comments and suggestions that have helped us to improve the quality of this manuscript. This document contains answers to the questions and comments by the reviewer. There are two parts of this document. The first part answers the remarks in the ‘Review Report Form’ of the review report. The second part contains answer to questions by the reviewer in the section, ‘Comments and Suggestions for Authors’ of the review report.
Answers to Comments from Reviewer 3
Comment 1:
Fig. 1: Provide some test for r in values where the logistic maps present non chaos (around 3.82, for instance).
Answer to Comment 1:
Thank you so much for your comments. The comments really assisted in enhancing the quality of the work. I followed all the valuable comments and modified the manuscript as per your valuable suggestion. Respected reviewer as per your question here we are interested in chaos for which we have added a bifurcation diagram and we are not interested in the non-chaotic region. Chaotic regions show high randomness in the chaotic region that we utilized for the image encryption process. The existence of a Bifurcation diagram is not possible for non-chaotic regions. The specified (3.82) is a chaotic region. The non-chaotic region is above 4 or less than 3.74. So we are not interested in the non-chaotic region and we are interested in the chaotic region.
Comment 2:
line 160: Chaotic parameters: define the reliable values for r and beta.
Answer to Comment 2:
Thank you so much for your comments. The comments really assisted me in enhancing the quality of the work. The work has been substantially enhanced in the modified manuscript version as per your valuable suggestions. Respected reviewer, we set the chaotic parameters as shown in Line 181 in the modified version of the manuscript.
Comment 3:
line 172: step 3 should be clarified. How is it done? d) How is Table 1 defined?
Answer to Comment 3:
Line 172 is now Line 193 (Point 3) in the modified manuscript and clarified it as per your valuable suggestions. The step is modified to (Select unique 256 (ranging from 0 to 255) elements from array obtained from step 2.)
Comment 4:
Image quality measures should be compared or given some analysis to check if the values are good or not.
Answer to Comment 4:
Thank you so much for your valuable suggestion that helped me a lot in enhancing the quality of the work. It wouldn’t be possible without your proper assistance and valuable suggestions. You have provided a way more convenience that can’t be ignored at any level.
I applied various statistical tests on the proposed scheme. However, as per your comment, I haven’t included references in the previous manuscript. I compared all the results in the modified version of the manuscript. Please see Table 6.
Comment 5:
line 311: there is no estimation on the key space. The authors present only a general perspective and qualitative information
Answer to Comment 5:
The keyspace assessment section is modified and added and check key sensitivity test. The system is examined with the original key and wrong decryption key. The decryption is failed with the wrong decryption key as shown in Table 7 of the modified version of the manuscript. The system is tested with original correct keys which are
Comment 6:
Some computational cost should be provided, as one of the key features of the paper is to reduce the cost
Answer to Comment 6:
Time Complexity Analysis is added to the manuscript. Minimum computational cost and resources should be used for an efficient encryption algorithm. We have computed the time taken to encrypt each image by using MATLAB 19. We have also listed some comparative analyses in Table 7. As decryption of the algorithm is the reverse of encryption therefore time utilized for decryption is the same. In comparison with existing schemes, it is seen from Table 8 that the proposed scheme has less computational complexity.
Comment 7:
The authors should also present some tests to lost of information. Please refer to the following reference: “Image encryption based on the pseudo-orbits from 1D chaotic map,” Chaos: An Interdisciplinary Journal of Nonlinear Science. Available: http://aip.scitation.org/doi/10.1063/1.5099261
Answer to Comment 7:
Thank you so much for the constructive comments/feedback. The aforementioned article is relevant to my article. I have cited the suggested article in my manuscript. Please see reference 38. I have added some other papers as well to enhance the quality of the work.
Comment 8:
Please, stress what is the novelty of the paper compared with other related manuscripts
Answer to Comment 8:
Previously many researchers and cryptographers have designed schemes that were using simple classical one and two-dimensional chaotic maps to achieve confusion and diffusion properties introduced by Claude Shannon. Many of the schemes are so simple that are utilized for the image encryption process. Khan et al [https://link.springer.com/article/10.1007/s11042-019-07818-4] in his recent published article appropriated the shuffling and substitution process using Henon chaotic map circle chaotic map and duffing chaotic map. Comparative to this we utilized complex quantum systems which is a modified version of traditional chaotic maps. We utilized S-box and compared it to the results of existing schemes as well. The computed results depicted that the anticipated scheme is highly secure. Younas et. al [https://www.mdpi.com/1099-4300/20/12/913] computed the result for the proposed scheme is too low compared to our proposed scheme. This makes our scheme more superior to existing schemes.
Comment 9:
2.2 Minor comments: a) l. 28: Revise the sentence: “A wide range of physical happenings can be prescribed by utilizing mathematical models.”
Answer to Comment 9:
L 28: The work is substantially revised as per your valuable suggestions. Please see line 28 to 30 in the revised manuscript. Minor changes are marked red in the manuscript. Additionally, I have added track changes for your convenience.
Comment 10:
It would also improve the paper if the figure captions would be made more self-contained. In addition to what is shown for which parameter values, one could also consider a sentence or two saying what is the main message of each figure.
Answer to Comment 10:
Thank you so much for your valuable suggestions. I have modified the manuscript and captions are corrected and explained in each paragraph.

Round 2
Reviewer 1 Report
The contribution of the proposed work is still not sufficient and the authors have not answered all the important questions.
Author Response
Answers to Remarks, Comments, and Suggestions -- Reviewer 1 (Round 2)
General:
We are extremely thankful to the anonymous reviewer for the useful comments and suggestions that have helped us to improve the quality of this manuscript. This document contains answers to the questions and comments by the reviewer. The file contains answers to questions by the reviewer in the section, ‘Comments and Suggestions for Authors’ of the review report.
Answers to Comments from Reviewer 1 (Round 2)
Comment 1:
As the main contribution of the paper is the design of a new s-box based on the Lucas sequence. Then, it is necessary, first to justify the use of Lucas sequence and especially to verify the robustness of the proposed s-box by applying some tests: bijectivity, non-linearity, strict avalanche criterion (SAC), output bits’ independence criterion (BIC), and equiprobable input/output XOR distribution. After that, you can compare the performance of the proposed s-box with the performance of other published S-boxes.
Answer to Comment 1:
Thank you so much for your comments. The comments really helped me enhancing the quality of the work. The work has been substantially enhanced in the modified manuscript as per your valuable suggestions.
We have added all the important tests of the S-box to the manuscript. Please see section 5 (Performance Analysis) where we added subsection 5.1 (Robustness of Proposed S-box). The assessments here include non-linearity test analysis (NL), strict avalanche criterion (SAC), bit Independence criterion (BIC), and differential approximation probability (DP). The result obtained using the proposed S-box is compared to the existing S-boxes. Each test result depicts that the constructed S-box for the encryption model has a better ability to resist any attack. Please see line 275 to 324, we have tabulated six different Tables and compared our result to existing schemes. The work is more detailed in Reference [41]. However, we have compared results to six different S-boxes [Please see Ref: 42-47].
We have modified our introduction part as well and added some more information relevant to quantum chaos. Please see (line 31 to 54), (line 62 to 66), and (line 89 to 97). We have revised all references as well to the best of our knowledge. Typos mistakes are thoroughly checked.
Section-wise:
Comment 2:
Line 21 replace efficiency with efficiency in terms of statistical analysis.
Answer to Comment 2:
Thank you so much for your comments. The comments really helped me enhancing the quality of the work. The work is substantially enhanced as per your valuable suggestions in the modified version of the manuscript. Respected reviewer, I corrected the sentence. Please see Line 21.
- A. Line 63: replace diffusion by confusion.
Answer: The work is thoroughly checked for grammatical mistakes and other technical issues. We changed the word diffusion with confusion. Please see line 99 in the newly modified manuscript. Other grammatical mistakes and other technical issues are also corrected. Thank you so much.
- B. You must mention an operation of quantization allowing to have values between 0 and 255, from values 0 and 1.
Answer: Thank you so much I changed the work accordingly as per your suggestions. Please see line 176 to 177.
- C. Line 142, replace:
This approach focused on two concepts, chaotic quantum map, and substitution box.
By
The system has two layers: a diffusion layer using a chaotic quantum map and a confusion layer using an s-box (as substitution operation).
Answer: The work is modified as per your suggestions. Please see line 199 to 201.
- D. It is necessary to detail how you build the inverse s-box.
You cannot obtain an inverse chaotic system for your chaotic system (in the decryption side, you use the same chaotic system in your cryptosystem).
Answer: Thank you so much for your valuable time. The question is interesting. We can invert our S-box by replacing input with output.
- In formula (5), replace logarithm in base e by logarithm in base 2
Answer: Thank you so much. The work is substantially changed apart from your comments. I replaced logarithm in base e with logarithm in base 2. Please see Line 371.

Reviewer 2 Report
The manuscript has been significantly improved.
Author Response
Answers to Remarks, Comments and Suggestions -- Reviewer 2 (Round 2)
General:
We are extremely thankful to the anonymous reviewer for the useful comments and suggestions that have helped us to improve the quality of this manuscript. This document contains answers to the questions and comments by the reviewer. The file contains answers to questions by the reviewer in the section, ‘Comments and Suggestions for Authors’ of the review report.
Answers to Comments from Reviewer 2 (Round 2)
Comment:
The manuscript has been significantly improved.
Answer to Comment :
Thank you so much for your positive comment that you are happy with our modified work. We have removed all typos mistakes as well and added some information on quantum theory to the introduction part (lines 31 to 54, 61 to 66, 89 to 97). Apart from this, we have added S-box tests to section 5. Please see blue marked modified work (line 275-324). Multiple tests are six tables are added and all the results are compared to existing S-box results.

Reviewer 3 Report
The authors have replied properly my questions.
Author Response
Answers to Remarks, Comments, and Suggestions –Reviewer 3 (Round 2)
General:
We are extremely thankful to the anonymous reviewer for the useful comments and suggestions that have helped us to improve the quality of this manuscript. This document contains answers to the questions and comments by the reviewer. There are two parts to this document. The first part answers the remarks in the ‘Review Report Form’ of the review report. The second part contains answers to questions by the reviewer in the section, ‘Comments and Suggestions for Authors’ of the review report.
Answers to Comments from Reviewer 3 (Round 2)
Comment 1:
The authors have replied properly to my questions.
The introduction part can be improved.
Answer to Comment 1:
Thank you so much for your positive comment that you are happy with our modified work. We have removed all typos mistakes as well and added some information on quantum theory to the introduction part (lines 31 to 54, 61 to 66, 89 to 97). Apart from this, we have added S-box tests to section 5. Please see blue marked modified work (line 275-324). Multiple tests are six tables are added and all the results are compared to existing S-box results. References and typos are thoroughly checked. Thank you once again for your valuable suggestions. The manuscript is blue highlighted in the second round with track changes.
